# IF1 Promotes Cellular Proliferation and Inhibits Oxidative Phosphorylation in Mouse Embryonic Fibroblasts under Normoxia and Hypoxia

**DOI:** 10.3390/cells13060551

**Published:** 2024-03-21

**Authors:** Lothar Lauterboeck, Sung Wook Kang, Donnell White, Rong Bao, Parnia Mobasheran, Qinglin Yang

**Affiliations:** 1Cardiovascular Center of Excellence, Louisiana State University Health Sciences Center, New Orleans, LA 70112, USA; lothar.lauterboeck@thermofisher.com (L.L.); sungwook.kang@bcm.edu (S.W.K.); dwhi20@lsuhsc.edu (D.W.III); 00008584@whu.edu.cn (R.B.); pmobass@lsuhsc.edu (P.M.); 2Cell Biology, Life Science Solutions, Thermo Fisher Scientific, Frederick, MD 21704, USA; 3Department of Pharmacology and Experimental Therapeutics, School of Graduate Studies, Louisiana State University Health Sciences Center, New Orleans, LA 70112, USA; 4School of Medicine, Louisiana State University Health Sciences Center, New Orleans, LA 70112, USA

**Keywords:** ATP synthase inhibitory factor subunit 1, cellular respiration, mitochondria, cellular glycolysis, cell proliferation, cellular viability, ATP, mouse embryonic fibroblasts

## Abstract

ATP synthase inhibitory factor subunit 1 (IF1) is an inhibitory subunit of mitochondrial ATP synthase, playing a crucial role in regulating mitochondrial respiration and energetics. It is well-established that IF1 interacts with the F1 sector of ATP synthase to inhibit the reversal rotation and, thus, ATP hydrolysis. Recent evidence supports that IF1 also inhibits forward rotation or the ATP synthesis activity. Adding to the complexity, IF1 may also facilitate mitophagy and cristae formation. The implications of these complex actions of IF1 for cellular function remain obscure. In the present study, we found that IF1 expression was markedly upregulated in hypoxic MEFs relative to normoxic MEFs. We investigate how IF1 affects cellular growth and function in cultured mouse embryonic fibroblasts derived from mouse lines with systemic IF1 overexpression and knockout under normoxia and hypoxia. Cell survival and proliferation analyses revealed that IF1 overexpression exerted limited effects on cellular viability but substantially increased proliferation under normoxia, whereas it facilitated both cellular viability and proliferation under hypoxia. The absence of IF1 may have a pro-survival effect but not a proliferative one in both normoxia and hypoxia. Cellular bioenergetic analyses revealed that IF1 suppressed cellular respiration when subjected to normoxia and was even more pronounced when subjected to hypoxia with increased mitochondrial ATP production. In contrast, IF1 knockout MEFs showed markedly increased cellular respiration under both normoxia and hypoxia with little change in mitochondrial ATP. Glycolytic stress assay revealed that IF1 overexpression modestly increased glycolysis in normoxia and hypoxia. Interestingly, the absence of IF1 in MEFs led to substantial increases in glycolysis. Therefore, we conclude that IF1 mainly inhibits cellular respiration and enhances cellular glycolysis to preserve mitochondrial ATP. On the other hand, IF1 deletion can significantly facilitate cellular respiration and glycolysis without leading to mitochondrial ATP deficit.

## 1. Introduction

Mitochondrial ATP synthase (F_1_F_o_-ATP synthase or complex V) is one of the most essential complexes in mammalian mitochondria for adenosine triphosphate (ATP) production in cells. The proton motive force (PMF) derived from the electron transport chain (ETC) across the mitochondrial membrane drives the forward rotation (bottom view) of the ATP synthase to catalyze adenosine diphosphate (ADP) to ATP. Under conditions such as hypoxia and mitochondrial membrane depolarization due to pathological stresses, the reverse rotation of ATP synthase can prevent the collapse of mitochondrial membrane potential, the primary trigger of mitochondrial permeability transition pore (mPTP) opening and apoptotic/necrotic cell death. ATP synthase inhibitory factor subunit 1 (ATPIF1 or IF1) was discovered 60 years ago [1]. In vitro assays using purified IF1 from bovine and submitochondria/ATP synthase in this study suggested the role of IF1 as an ATP synthase-interacting protein that inhibits ATP hydrolysis [1]. Moreover, IF1 interacts with the F1 domain of ATP synthase when mitochondria become depolarized under pathological conditions, such as hypoxia [2,3]. The interaction of IF1 with the inhibitory state of ATP synthase has been well-characterized using cryo-electron microscopes [4,5]. However, it remains unclear how hypoxia may impact the effects of IF1 on cellular respiration and energetics.

Most early studies support that IF1 can bind to ATP synthase to inhibit ATP hydrolysis [6,7]. Whether IF1 also inhibits the ATP synthetic activity of ATP synthase remains controversial. However, most early studies were based on biochemical assays on isolated submitochondrial particles with no intact mitochondrial intramembrane. Evidence supporting IF1’s inhibitory effects on ATP synthesis has been emerging. IF1 overexpression blocks the synthetic activities of ATP synthase in primary cultured neurons [8]. Since cellular respiration is closely related to ATP synthase activity, inhibiting ATP synthase in both directions should reduce the rate of oxidative phosphorylation (OXPHOS). Investigations assessing cellular respiration rates in cultured cells, such as INS-1 beta-cells [9] and rat neonatal cardiomyocytes [10], revealed that IF1 is inversely related to cellular respiration. We previously found that cellular energetics were markedly increased in SiRNA-mediated IF1 knockdown, consistent with the earlier finding, and suppressed in adenovirus-mediated IF1 overexpression in cultured INS-1 cells [11]. However, the nature of this inverse relationship remains incompletely established.

Cellular energetics plays a critical role in regulating cellular survival and proliferation. IF1 is upregulated in many tumors and cancer cells, which may promote cancer cell survival [12,13,14,15]. Eliminating IF1 in various cancer cell lines modified cellular bioenergetics and decreased migration, invasion, and proliferation [16,17]. However, how cellular changes of IF1 impact cellular energetics, cell viability, and proliferation remains poorly understood. Most previous studies on the role of IF1 either used in vitro-isolated submitochondria combined with purified proteins or cultured cells with contrived overexpression of IF1, its mutants, or RNAi. Investigations using cultured cells derived from transgenic IF1 KO or overexpression animals may help shed new light on the effects of IF1 on cellular respiration and bioenergetics.

In the present study, we investigate the effects of IF1 overexpression and knockout in mouse embryonic fibroblasts (MEFs) on cellular viability, proliferation, bioenergetics, and glycolysis when subjected to normoxia and hypoxia. We found that IF1 is inversely related to cellular respiration and bioenergetics, with relatively modest effects on cellular survival and proliferation.

## 2. Materials and Methods

### 2.1. Animals

IF1^−/−^ Mice: Details of the IF1^−/−^ mouse line were reported previously [18,19,20]. The mouse strain has been maintained in the C57/B6J background.

IF1 Overexpression (IF1 OE) Mice: A transgenic line with global IF1 overexpression was established using a previously described approach [21,22,23] and was described in our previous publication [11].

Animals received food and water on *ad libitum* basis, and lighting was maintained on a 12-h cycle. All experimental procedures were conducted by the Guide for Care and Use of Laboratory Animals of the National Institutes of Health and were approved by the Institutional Animal Care and Use Committee of the University of Alabama at Birmingham and Louisiana State University Health Science Center-New Orleans.

### 2.2. Isolation of Mouse Embryonic Fibroblasts

Mouse embryonic fibroblasts (MEFs) were isolated as described [24] with minor modifications. Embryos at embryonic day 13.5 were extracted from the uterine horn of pregnant mice, washed in phosphate-buffered saline (PBS, Gibco, Life Technologies, Waltham, MA, USA), and transferred to a new PBS solution. The heads and red organs were removed and the remaining tissue was transferred into a fresh petri dish (Nunc, Thermo Fisher, Waltham, MA, USA) containing 10 mL of 0.05% trypsin/ethylenediaminetetraacetic acid (EDTA, Gibco, Life Technologies, Waltham, MA, USA). Embryos were minced with scissors, mechanically dispersed, and incubated for 10 min at 37 °C. The digestion was stopped by the addition of 20 mL MEF medium consisting of Dulbecco’s Modified Eagle Medium (DMEM, Gibco, Life Technologies, Waltham, MA, USA), 10% fetal bovine serum (FBS, Atlanta Biologicals, Atlanta, GA, USA), and penicillin/streptomycin (Pen/Strep, Gibco, Life Technologies, Waltham, MA, USA). The suspension was centrifuged for 8 min at 200× *g*. The resultant supernatant was aspirated, resuspended, and plated in 100 mm Petri dishes. After they reached 100% confluency, cells were split and grown again until confluency. Afterward, cells were frozen in DMEM containing 15% FBS and 10% dimethyl sulfoxide (DMSO, Sigma-Aldrich, St. Louis, MO, USA) using a Mr. Frosty™ (cooling rate 1 °C/min, Nagle Nunc, Life Technologies, Waltham, MA, USA). Only passages between 3 and 6 were used for experiments.

### 2.3. Cell Culture

Cells were grown in DMEM containing 10% FBS and 1% pen/strep in a humidified incubator at 37 °C. At 80% confluency, cells were passed by trypsinization (0.05% trypsin/EDTA) for 5 min. To count the cells, they were centrifugated and resuspended in a 19 mL culture medium. An aliquot was removed, the same amount of Trypan blue was added, and 10 µL was analyzed in a hematocytometer. Cells were resuspended to a concentration of 1 × 10^6^ cells/mL and either plated for further growth or for experiments, as described below.

MTT cell viability assays: MEFs (10,000) were seeded into a 96-well plate (Corning, New York, NY, USA) and incubated overnight at 37 °C. The next day, the medium was changed to a culture medium supplemented with 1% 3-[4,5-Dimethylthiazol-2-yl]-2,5-diphenyltetrazolium bromide (MTT, Sigma-Aldrich, St. Louis, MO, USA). After 4 h, the medium was exchanged for lysis buffer (isopropanol, 0.1 M HCL, and 10% Triton X, obtained from Sigma-Aldrich, St. Louis, MO, USA). Absorbance was read at 550 nm vs. 650 nm reference using a Biotek Synergy HT plate reader (Biotek Instruments Inc., Winooski, VT, USA). For the Trypan blue exclusion assay, 100,000 cells were seeded into 24 well plates (Corning, New York, NY, USA)) and incubated overnight. For four days, cells were washed in PBS, trypsinized, and counted by mixing the same amount of Trypan blue (MP Biomedical LLC, Santa Ana, CA, USA) with the cell suspension. Cells were incubated in a C-shuttle (Biospherix, Parish, NY, USA) at 2% O_2_ inside the previously mentioned incubator for the same period as in normal oxygen to mimic oxygen deprivation (hypoxia). Each cell type was run in triplicates in three independent experiments.

### 2.4. Determination of Proliferation during Normoxia and Hypoxia

Cell proliferation was measured using Ki-67 and BrdU incorporation. MEFs were seeded into 8-well chamber slides (Lab-TekII, Nunc, Life Technologies, Waltham, MA, USA) and incubated overnight to assess proliferation. Then, 10 μM BrdU was added to the culture medium and incubated for 24 h in normoxia or hypoxia. Cells were fixed in 4% paraformaldehyde (PFA) for 15 min, followed by PBS washing. Cells were treated for 30 min with 2N hydrochloric acid (HCL, Sigma-Aldrich, St. Louis, MO, USA) to fluorescently label the incorporated BrdU at room temperature. After washing, the cells were incubated with a blocking solution (5% bovine serum albumin (BSA) in PBS with 0.1% Triton X) for 1 h at room temperature. The primary antibody (1:50, BrdU, Invitrogen, Waltham, MA, USA) was conjugated to Alexa Fluor 570 nm and was incubated overnight at 4 °C. The primary antibody was diluted in PBS containing 5% BSA and 0.1% Triton X. After washing, cells were incubated with 3 μM Hoechst (Invitrogen, Waltham, MA, USA) for 10 min and then imaged using Cytation 5 (Biotek Instruments Inc., Winooski, VT, USA). Four fields per well were taken for data analysis, and at least 20 cells per field were counted for colocalization of BrdU and Hoechst.

Ki-67 staining was facilitated similarly to BrdU staining. After cell seeding, cells were incubated for 24 h in either normoxia or hypoxia. Cells were fixed with 4% PFA for 15 min followed by washing with PBS. Cell membrane permeabilization and epitope blocking were conducted by incubating 0.3% Triton X and 5% BSA in PBS for 1 h, followed by PBS washing. The primary antibody (Ki-67 (SolA1 clone), 1:300, conjugated to Alexa Fluor 570 nm (Invitrogen, Waltham, MA, USA), was incubated overnight at 4 °C. After washing, cells were counterstained with 3 μM Hoechst and imaged. Four fields per well were taken for data analysis, and at least 20 cells per field were counted for colocalization of Ki-67 and Hoechst. Proliferation was calculated using the following equation: Hoechst-positive cells/Ki-67-positive cells × 100.

### 2.5. Cellular Bioenergetics

MEFs were seeded in a 24-well Seahorse XF^e^24 cell culture plate (Agilent, Santa Clara, CA, USA) at a concentration of 50,000 cells/well in 250 µL culture medium and incubated overnight at 37 °C and 5% CO_2_. Additionally, a low oxygen environment (2% O_2_) mimics hypoxia. Cells were exposed to hypoxia for 3 h. Cells were washed with either mitochondrial stress test (MST) or glycolysis stress test (GST) assay medium to remove the remaining FBS and covered with 450 µL assay medium. Afterward, the plate was incubated for 1 h in a CO_2_-free incubator at 37 °C. In the case of hypoxia, degassing also took place in a low O_2_ environment. The measurement of all plates took place in a normoxic environment. For the GST, the medium was composed of Seahorse XF base medium (Agilent) without phenol red supplemented with 5 mM 4-(2-hydroxyethyl)-1-piperazine ethane sulfonic acid (HEPES, Sigma-Aldrich, St. Louis, MO, USA), pH 7.4 (±0.04). For the mitochondrial stress test (MST), 1 mM sodium pyruvate (Sigma), 2 mM L-glutamine (Sigma), and 10 mM glucose (Sigma-Aldrich, St. Louis, MO, USA) were added to the GST medium.

While the medium was degassing, the injection ports of the cartridge were loaded with assay-specific drugs. The sequence and final concentrations for the MST were: 1.5 μM oligomycin (oligo, port A, Sigma-Aldrich, St. Louis, MO, USA), 3 μM carbonyl-cyanide-p-trifluoromethoxy phenylhydrazone (FCCP, port B, Sigma-Aldrich, St. Louis, MO, USA), and 1 μM antimycin A/0.5 μM rotenone (AA/Rot) in port C. Oligomycin was used to inhibit the mitochondrial complex V, and to measure the maximal respiration, FCCP was used. To determine the minimal amount of oxygen used by the cells, cellular respiration was inhibited by a combination of a complex I (rotenone) and complex III (antimycin A) inhibitor. For the GST assay, ports were loaded with 10 mM glucose (Glu, Sigma-Aldrich, St. Louis, MO, USA), 1.5 μM oligomycin, and 50 mM 2-Deoxy-D-glucose (2-DG, Sigma-Aldrich, St. Louis, MO, USA). The last injection also included 5μM Hoechst (Invitrogen, Life Technologies, Waltham, MA, USA) for cell normalization. The injection volumes were 50 μL for port A, 55 μL for B, and 60 μL for C. Each drug was optimized to achieve optimal conditions. The measuring cycle after injection consisted of 3 min of mixing, 2 min of waiting, and 3 min of measuring. A total of three cycles per injection was used. OCR (oxygen consumption rate) and ECAR (extracellular acidification rate) were normalized to cell number using Agilent imager software (Wave 2.6.1.56, 2018) and Cytation 5 (Gen5, 2018) by counting Hoechst-positive cells. OCR and ECAR values were normalized to 1000 cells. Three independent runs were performed using at least triplicates per genotype. Data were analyzed by the Agilent report generator program (Agilent, Santa Clara, CA, USA).

### 2.6. Lactate Determination

Lactate was analyzed under normoxic culture conditions and after exposure to 3 h hypoxia using the lactate colorimetric assay kit (Biovision, Milpitas, CA, USA) following the manufacturer’s instructions. Briefly, 100,000 cells were seeded into 24-well plates and incubated as used for bioenergetic evaluation. The supernatant was removed and assayed as described the manufacturer’s instructions. The values were normalized to total protein concentration.

### 2.7. Statistical Analysis

All experiments were performed in at least three independent experiments and, if applicable, in technical triplicates. The data are reported as mean ± SEM. Statistical analysis was performed using an unpaired Student’s *t*-test with Welch’s correction, one-way or two-way ANOVA (GraphPad Prism v.9.0, GraphPad Software Inc., La Jolla, CA, USA). A statistical significance was reached when *p* < 0.05. For data analysis, four fields per well were taken, and at least 20 cells per field were counted for colocalization of either BrdU or KI-67 with Hoechst.

## 3. Results

### 3.1. IF1 Plays a Role in Regulating Cell Survival and Proliferation, Especially under Hypoxia

MEFs derived from the global IF1 OE, IF1^−/−^ and WT mice were cultured under normoxia and hypoxia. Both IF1 OE and IF1^−/−^ mice show no phenotype during their first years of life under physiological conditions. MEFs from these two mouse lines are healthy, with similar morphological properties under culture conditions. A western blot was performed to validate the presence of IF1 overexpression and IF1^−/−^ in the respective MEFs (Appendix A). While IF1 OE MEFs showed a three-fold increase in IF1, IF1^−/−^ MEFs present no IF1 (Appendix A). To determine if IF1 expression changes in response to hypoxia, we performed real-time (RT)-qPCR on samples from controlled MEFs subjected to hypoxia. IF1 transcript was markedly increased (~three-fold) in hypoxic MEFs (Appendix A).

Under normoxia, cell count based on the Trypan blue assay indicates that all MEFs were similarly increased during the first 72 h (Figure 1A,B). IF1 OE MEFs were markedly increased at 72 h, and IF1^−/−^ slightly increased compared with WT MEFs (Figure 1A,B). To understand how IF1 influences cell viability and proliferation in normoxia, we first performed an MTT assay to assess cellular viability. IF1 OE and IF1^−/−^ MEFs showed no substantial changes during the 72 h of culture (Figure 1C,D). We next conducted a BrdU uptake assay on the above MESFs to investigate how IF1 affects MEF proliferation. Representative BrdU-positive and Hoechst nuclei images were shown (Figure 1E). The BrdU- to Hoechst-positive cell ratio in IF1 OE MEFs was modestly increased but unchanged in IF1^−/−^ MEFs (Figure 1F,G). We also conducted KI-67 fluorescent staining on the above cells to analyze their proliferation further (Figure 1H). Consistent with the BrdU uptake assay findings, positive Ki67 staining was increased in IF1 OE but unchanged in IF1^−/−^ MEFs (Figure 1I,J). We then assessed MEFs subjected to hypoxia. While cell numbers of WT MEFs steadily declined (Figure 2A,B) during the three-day culture, cell numbers of IF1 OE and IF1^−/−^ MEFs increased during the first 24 h and maintained the same levels during the next 48 h (Figure 2A,B). MTT cellular viability assays revealed that IF1 OE and IF1^−/−^ MEFs increased compared with WT MEFs (Figure 2C,D). Cellular proliferation evaluated using BrdU and Ki67 staining showed increased proliferation in cultured IF1 OE but no changes in IF1^−/−^ MEFs compared with WT MEFs (Figure 2E–J). These results suggest that IF1 exerts protective and proliferative actions, and IF1 knockout exerts protective action on cultured MEFs when subjected to hypoxia.

### 3.2. Effects of IF1 on Mitochondrial ATP Content under Normoxia and Hypoxia

It has been well documented that IF1 exerts its inhibitory effects on ATP synthase via direct interaction. We transfected the MEFs with a fluorescence-based mitochondria-specific ATP indicator (MaLionR) [25,26] to evaluate the effects of IF1 on mitochondrial ATP as previously described under both normoxia and hypoxia (Figure 3A. The results indicate that the intensities of mitochondrial ATP signals were unchanged among the three MEFs (Figure 3B) in normoxia. Nonetheless, when MEFs were subjected to hypoxia, the mitochondrial ATP signal was robustly increased in IF1 OE relative to WT MEFs (Figure 3B). In contrast, the mitochondrial ATP signal was reduced by about 50% in IF1^−/−^ MEFs compared with WT MEFs (Figure 3B). These results support the role of IF1 in preserving mitochondrial ATP.

### 3.3. Effects of IF1 on Mitochondrial Respiration in MEFs Subjected to Normoxia and Hypoxia/Reoxygenation

We first conducted mitochondrial stress assays to assess cellular respiration in IF1 OE, IF1^−/−^, and WT MEFs under normoxic conditions using the Seahorse Bioanalyzer (Agilent). It appears that IF1 overexpression suppresses cellular respiration in normoxia (Figure 4A). Specifically, basal and maximal respiration were both decreased in IF1 OE MEFs (Figure 4B,C) without affecting ATP production rate and proton leak rate (Figure 4D,E). However, the spare respiratory capacity was decreased in IF1 OE relative to WT MEFs (Figure 4F). On the other hand, cellular respiration was elevated in IF1^−/−^ relative to WT MEFs (Figure 4G). Except for spare respiratory capacity, all other respiration parameters, including basal, maximal, ATP production, and proton leaks, were upregulated (Figure 4H–L). We then further accessed the effects of IF1 on cellular respiration after a short period of hypoxia (3 h). While no changes could be detected in basal respiration (Figure 5B), the IF1 OE MEFs showed a decreased maximal and spare respiratory capacity compared with WT MEFs (Figure 5C,D). Interestingly, the ATP production rate was increased (Figure 5E), and protein leak was decreased in IF1 OE relative to WT MEFs (Figure 5E,F). In contrast, cellular respiration remained robustly upregulated in IF1^−/−^ MEFs after hypoxia compared with WT MEFs (Figure 5G). Most respiratory parameters, including basal, maximal, spare respiratory capacity, and proton leaks, were elevated in IF1^−/−^ relative to WT MEFs (Figure 5G–L). Interestingly, the ATP production rate was unchanged in IF1^−/−^ relative to WT MEFs (Figure 5K). These results indicate that IF1 impacts cellular respiration in normoxia and hypoxia/reoxygenation under cultured conditions.

### 3.4. Effects of IF1 on Cellular Glycolysis in MEFs Subjected to Normoxia and Hypoxia/Reoxygenation

To investigate how cellular glycolysis is changed in response to the impacts of IF1 on cellular respiration in MEFs subjected to normoxia and hypoxia/reoxygenation, we performed glycolysis stress assays using the Seahorse Bioanalyzer. Under hypoxia, the trace of the overall glycolysis stress in IF1 OE MEFs showed a modest change (Figure 6A) with no differences in glycolysis. Still, it decreased glycolytic capacity and reserve (Figure 6C,D). We also assessed the lactate content and confirmed unchanged glycolysis in IF1 OE relative to WT MEFs (Figure 6E). However, IF1 OE MEFs appear to have an increased impact on the overall glycolytic stress response (Figure 6F) with increased glycolysis and glycolytic capacity (Figure 6G,H) but unchanged glycolytic reserve (Figure 6I). Unexpectedly, lactate content was modestly reduced in IF1^−/−^ MEFs subjected to hypoxia/reoxygenation (Figure 6J). On the other hand, the glycolysis stress assay demonstrates increased glycolysis and glycolytic capacity rates but unchanged glycolytic reserve in IF1^−/−^ relative to WT MEFs in normoxia (Figure 7A–D). Lactate content was also increased in IF1^−/−^ relative to WT MEFs (Figure 7E). In MEFs subjected to hypoxia/reoxygenation, the absence of IF1 further augmented all parameters of the glycolysis stress assay, including glycolysis, glycolytic capacity, and glycolytic reserve (Figure 7F–I). Lactate content was also correspondingly increased in IF1^−/−^ relative to WT MEFs (Figure 7J). Therefore, it appears that both IF1 overexpression and IF1 KO impact cellular glycolysis in normoxia and hypoxia/reoxygenation probably in response to the changes in cellular energetic states.

## 4. Discussion

The mitochondrial ATP synthase produces the majority of ATP through OXPHOS in cells. IF1 has long been identified as an endogenous inhibitory protein via its insertion into the F1 sector [1,25,26,27]. However, the exact effects of IF1 on overall cellular respiration under normoxia and hypoxia remain obscure. The present study demonstrates that IF1 affects survival, proliferation, and cellular respiration, based on experiments using MEFs from IF1 OE and IF1^−/−^ mice compared to their corresponding WT controls.

While many investigations have demonstrated that IF1 promotes the proliferation of several cancer cells [13,14,15,16,27], how IF1 impacts the growth and viability of healthy cells remains unclear. Most of the previous studies on the role of IF1 in cultured cells were based on partial knockdowns or cell lines with intrinsically high IF1 expression. To remedy the above shortfalls, we carried out investigations using MEFs from a transgenic line with global IF1 OE [11] and a gene-targeting mouse line of IF1^−/−^ [11,18,19,20] to evaluate how a modest IF1 overexpression and complete IF1 knockout would affect cellular viability and proliferation, cellular respiration, cellular glycolysis, and mitochondrial ATP production.

Our results provide evidence supporting the previous findings that IF1 plays a regulatory role in cellular viability and proliferation. More importantly, our results demonstrate that the proliferative effects of IF1 are greatly enhanced when cells are subjected to hypoxia. These results appear consistent with most previous findings in cancer cells [13,14,15,16,27]. On the other hand, the absence of IF1 does not affect cell growth during normoxia. Interestingly, both IF1 overexpression and knockout exert pro-survival effects on cultured MEFs. How removal and/or overexpression of IF1 lead to the same pro-survival effects is an intriguing question. The same cellular effects of removal or OE of IF1 might be associated with compensatory effects of IF1 knockout. For instance, it has been discussed that some IF1 homologues, isoforms, or gene copies may exist in eukaryotes from yeast to animals and humans [28]. Removal of a single IF1 gene, as conducted in this study, may lead to compensatory expression of other IF1 copies or isoform genes that may be present in the eukaryotic chromosomes. This could explain why removal or overexpression of IF1 may lead to a similar phenotype in whole cells. The molecular mechanisms underlying IF1’s proliferative and pro-survival effects remain incompletely understood. It has been shown that IF1 is involved in regulating mitochondrial morphology through cristae formation by IF1 stabilization of the dimeric and oligomeric ATP synthases [4,29,30], so these pro-cristae formation effects of IF1 may contribute to its pro-survival effects.

Moreover, it has been suggested that IF1 may be involved in regulating mitochondrial morphology, cristae density, and mitophagy via indirect regulation of Opa1 [18]. However, we did not detect major changes in Opa1 levels either in IF1 OE or IF1^−/−^ MEFs (Appendix A); thus, it appears that IF1 exerts limited changes in mitochondrial ultrastructure in MEFs as compared to those in the liver [18]. A recent investigation suggested that IF1 interacts with the p53–cyclophilin D complex and promotes the opening of the permeability transition pore (mPTP) [31]. Still, our results do not support this occurring in cultured MEFs because hypoxia should have exacerbated cell death in hypoxic IF1 OE MEFs, partly due to increased mPTP opening.

On the other hand, ATP preservation is markedly improved in IF1 OE MEFs under hypoxia, and mitochondrial OXPHOS is improved in IF1^−/−^ MEFs under normoxia and hypoxia/reoxygenation. The IF1’s cellular protective role in hypoxia/reoxygenation appears consistent with the recent finding that kynurenic acid indirectly activates IF1 to protect cardiomyocytes from ischemia/reperfusion injury [32]. It has also been suggested that IF1 may exert protective effects via elevated ROS-related survival signals in the brain [8] and intestine [33]. IF1’s cellular protective role can be mediated by interacting with other subunits of ATP synthase or non-ATP synthase-related mechanisms. IF1 can protect cancer cells from apoptosis via binding to the oligomycin-sensitivity-conferring protein (OSCP) subunit of ATP synthase [13]. IF1 may promote pathological cardiac hypertrophy via cytosolic activation of calcium–calmodulin kinase II to disrupt mitochondrial calcium handling (CaMKII) [34]. Nonetheless, further evidence of cytosolic IF1 interactions and its non-ATP synthase-interacting effects remains scarce. Further investigations are needed to support potential non-ATP synthase-interacting effects of IF1. Our long-term observation of mouse models with IF1 overexpression and knockout implies that IF1 does not exert significant biological effects under basal conditions. It remains probable that the pro-survival effects of IF1 OE are via preserving ATP and improving cellular respiration.

The mitochondrial stress assay results revealed that IF1 overexpression modestly suppressed basal and maximal respiration when cells were subjected to normoxia and spared respiration capacity without impacting ATP production and proton leak. When cells were subjected to hypoxia/reoxygenation, IF1 overexpression further suppressed maximal respiration and spared respiration capacity with increased proton leak. These findings are largely consistent with previous results from us and other groups on different cell types under normoxia [10,11,35]. In IF1 knockout MEFs, all cellular respiration rates were markedly upregulated under both normoxia and hypoxia/reoxygenation. These results are somewhat similar to our previous finding in INS-1 cells and a previous finding that mitochondria from IF1 knockout hearts were resistant to impaired mitochondrial respiration due to pathological cardiac hypertrophy [10]. Nonetheless, another investigation on cultured HeLa cells with the overexpression of WT and a dominant negative mutant of IF1 revealed an opposite effect of IF1 on cellular respiration [18]. This discrepancy is likely related to cell types, culture conditions, and WT and mutant IF1 proportion relative to ATP synthase. It has been reported that the ratio of active ATP synthase and inactive (IF1-bound) ATP synthase pools may contribute to different cellular respiration [36]. It appears that when IF1 is primarily absent, cellular respiration tends to increase.

Under sub-optimal culture conditions, IF1 appears to show minor regulatory effects on cell survival and proliferation, as well as regulating OXPHOS and glycolysis. Under pathological conditions, these effects are amplified. It appears that IF1 exerts modest inhibition of cellular respiration and correspondingly upregulated glycolysis even in normoxia. These results seemingly contradict the previous finding that IF1 only associates with F_1_ when membrane depolarization occurs. Likely, the less dramatic change induced by the loss and gain of IF1 function is not optimal. However, this change does not seem sufficient to be reflected in the mitochondrial ATP content. In hypoxia, it is more apparent that IF1 exerts more inhibition on cellular respiration and leads to greater glycolytic ATP transfer from the cytosol. Our results of mitochondrial membrane potential measurement indicate that the absence of IF1 in MEFs subjected to normoxia and hypoxia did not lead to further mitochondrial depolarization, probably due to the import of ATP from increased cytosolic glycolysis in IF1^−/−^ MEFs. On the other hand, IF1 overexpression led to modest mitochondrial hyperpolarization (Appendix A), which may explain how IF1 OE MEFs preserve more ATP under hypoxia. This result also indirectly supports the idea that IF1 inhibits ATP synthase and leads to the build-up of protons, at least during the hypoxic period.

The present study provides in-depth insights into the role of IF1 in regulating cellular energetics. We found that the initial loss of ATP in MEFs without IF1 leads to a marked activation of the cellular glycolysis activity, probably due to the reduction in the ATP/ADP ratio [37]. This upregulation of glycolysis in IF1^−/−^ MEFs is even more dramatic when cells are subjected to hypoxia. Through the reverse transport mechanism [38,39], glycolytic ATP appears to supply the mitochondria sufficiently. Meanwhile, the reduced respiration rate in MEFs with IF1 overexpression also triggers a modest upregulation of glycolysis despite the preservation of mitochondrial ATP. The cellular respiration changes in response to hypoxic stress suggest that IF1 plays a more critical role in stopping the futile mitochondrial consumption of ATP. Combined with the direct visualization of mitochondrial ATP, our results at least indirectly indicate that IF1 may inhibit both ATP hydrolysis and synthesis. The long-lasting contradiction with findings from in vitro experiments using submitochondrial samples and structural biological findings may be the result of non-intact mitochondria, the chronic sum of impact instead of transient alterations, the degrees of IF1 ablations, and possible different IF1 actions in mitochondria from other species and cell types. Direct interaction of extra-mitochondrial IF1 with cell membrane ATP synthase remains elusive. So far, no evidence supports the existence of cell membrane ATP synthase in embryonic fibroblasts.

## 5. Conclusions

In summary, the present study demonstrates that IF1 plays a role in cellular respiration and indirectly in cellular glycolysis, especially when cells are subjected to pathological conditions, such as hypoxia. IF1 effects on cellular viability and proliferation should be the direct consequences of changing cellular respiration state and glycolysis.

## Figures and Tables

**Figure 1 cells-13-00551-f001:**
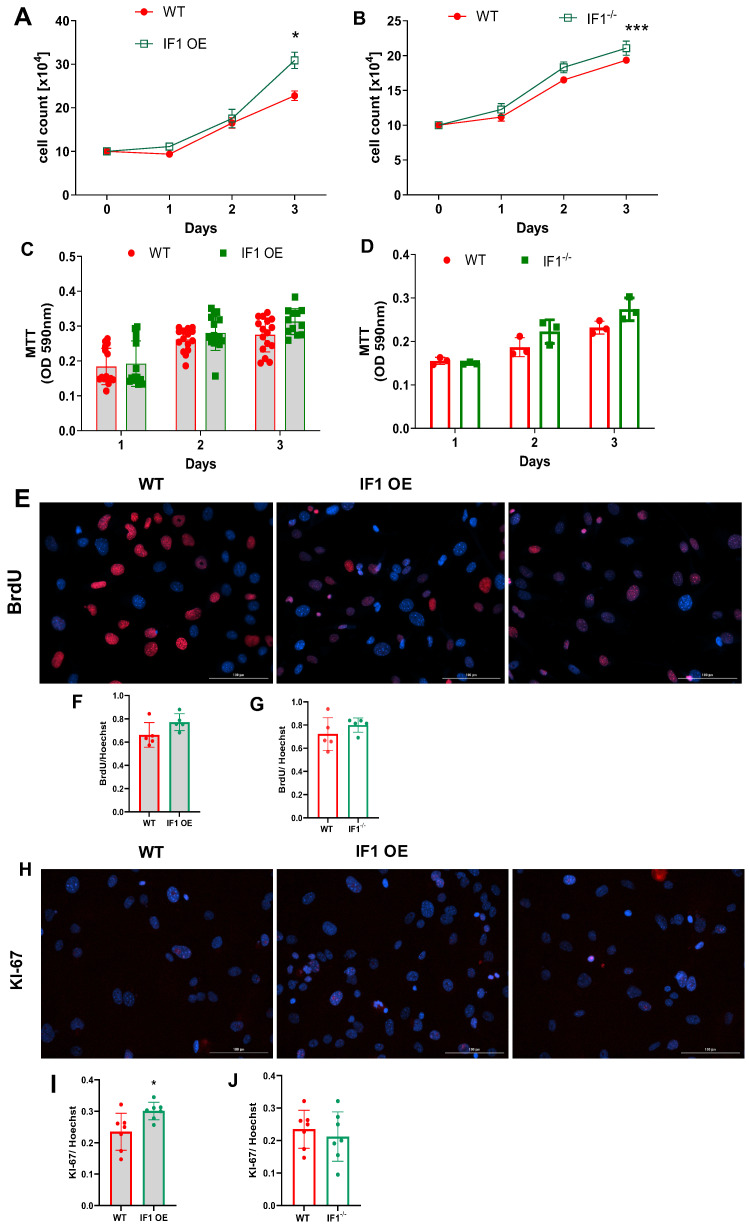
Effects of IF1 ablations on the proliferation and survival of the MEFs in normoxia. Cellular proliferation and survival of cultured IF1^−/−^ and IF1 OE MEFs using Trypan blue exclusion assay, MTT cell viability assays, immunostaining of BrdU and KI-67. (**A**) Trypan blue exclusion assay on IF1 OE and WT MEFs. (**B**) Trypan blue exclusion assay on IF1^−/−^ and WT MEFs. (**C**) MTT assays on IF1 OE and the respective WT MEFs. (**C**) Trypan blue exclusion assay on IF1^−/−^ and WT MEFs. (**D**) MTT cell viability assays on IF1^−/−^ and WT MEFs. (**E**) Fluorescent images of BrdU (red) and Hoechst (blue) on IF1 OE and WT MEFs under normoxia. (**F**) Quantification of BrdU to Hoechst ratio in WT and IF1 OE MEFs. (**G**) Quantification of BrdU to Hoechst ratio in WT and IF1^−/−^ MEFs. (**H**) Fluorescent images of Ki67 (green) and Hoechst (blue) on IF1 OE and WT MEFs under normoxia. (**I**) Quantifying Ki67 to Hoechst ratio in WT and IF1 OE MEFs. (**J**) Quantification of Ki67 to Hoechst ratio in WT and IF1^−/−^ MEFs. Data are presented as mean ± SEM of three independent experiments. For image data analysis, 4 fields per well were taken, and at least 20 cells per field were counted for colocalization of either BrdU or KI-67 with Hoechst. * *p* < 0.05, *** *p* < 0.001 when compared by two-way ANOVA followed by Tukey analysis and Student’s *t*-test.

**Figure 2 cells-13-00551-f002:**
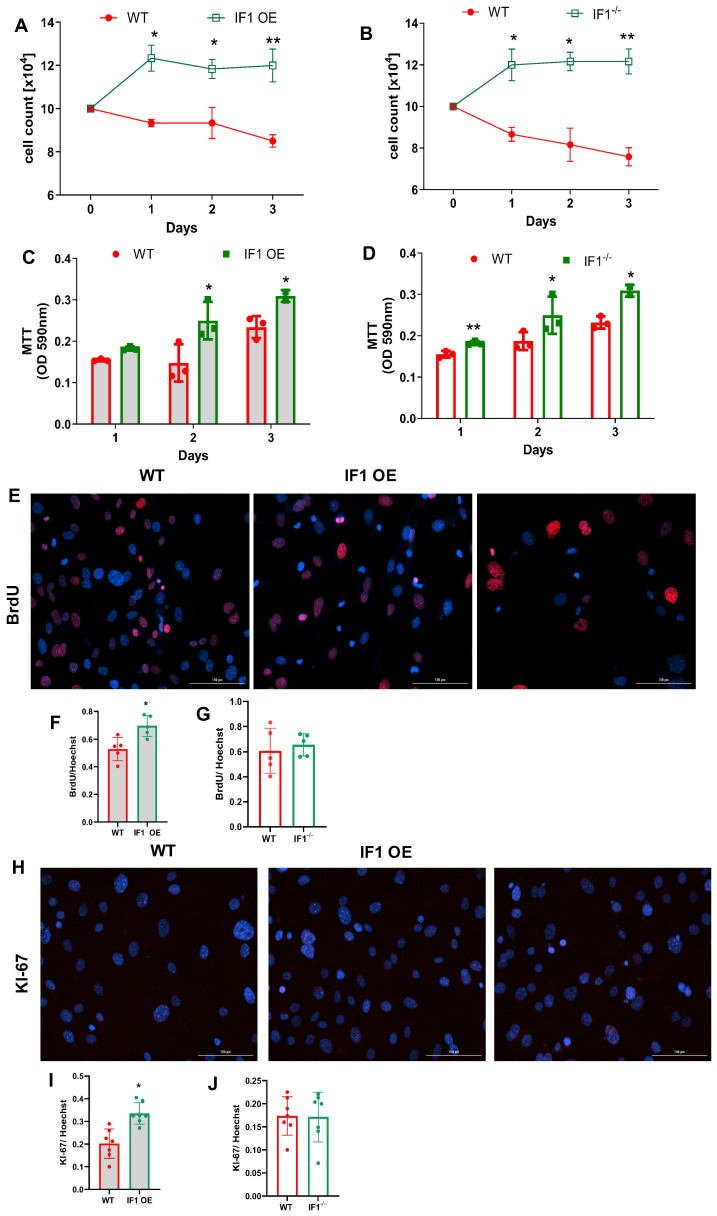
Effects of IF1 ablations on the proliferation and survival of the MEFs in hypoxia. Cellular proliferation and survival of cultured IF1^−/−^ and IF1 OE MEFs under the hypoxia using Trypan blue exclusion assay, MTT cell viability assays, and immunostaining of BrdU and KI-67. (**A**) Trypan blue exclusion assay on IF1 OE and WT MEFs. (**B**) Trypan blue exclusion assay on IF1^−/−^ and WT MEFs. (**C**) MTT assays on IF1 OE and the respective WT MEFs. (**C**) Trypan blue exclusion assay on IF1^−/−^ and WT MEFs. (**D**) MTT cell viability assays on IF1^−/−^ and WT MEFs. (**E**) Fluorescent images of BrdU (red) and Hoechst (blue) on IF1 OE and WT MEFs under hypoxia. (**F**) Quantification of BrdU to Hoechst ratio in WT and IF1 OE MEFs. (**G**) Quantification of BrdU to Hoechst ratio in WT and IF1^−/−^ MEFs. (**H**) Fluorescent images of Ki67 (green) and Hoechst (blue) on IF1 OE and WT MEFs under hypoxia. (**I**) Quantification of Ki67 to Hoechst ratio in WT and IF1 OE MEFs. (**J**) Quantification of Ki67 to Hoechst ratio in WT and IF1^−/−^ MEFs. Data are presented as mean ± SEM of three independent experiments. * *p* < 0.05, ** *p* < 0.01 when compared by two-way ANOVA followed by Tukey analysis and Student’s *t*-test.

**Figure 3 cells-13-00551-f003:**
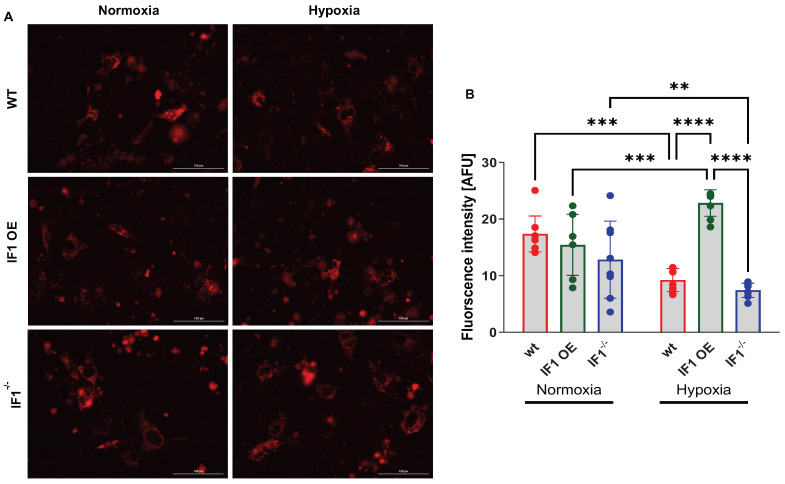
Real-time imaging and quantification of IF1 OE, IF1^−/−^ and WT MEFs with the expression of a fluorescent mitochondria-specific ATP-indicator (MaLionR). (**A**) Images of IF1 OE, IF1^−/−^ and WT MEFs with the MaLionR under normoxia and hypoxia. (**B**) Quantification of fluorescent mitochondrial ATP signal intensity in IF1 OE, IF1^−/−^ and WT MEFs in normoxia and hypoxia. Data are reported as mean ± SEM. *n* = 3 independent experiments. Data were analyzed using a two-way ANOVA followed by Tukey analysis. ** *p* < 0.01, *** *p* < 0.001, **** *p* < 0.0001.

**Figure 4 cells-13-00551-f004:**
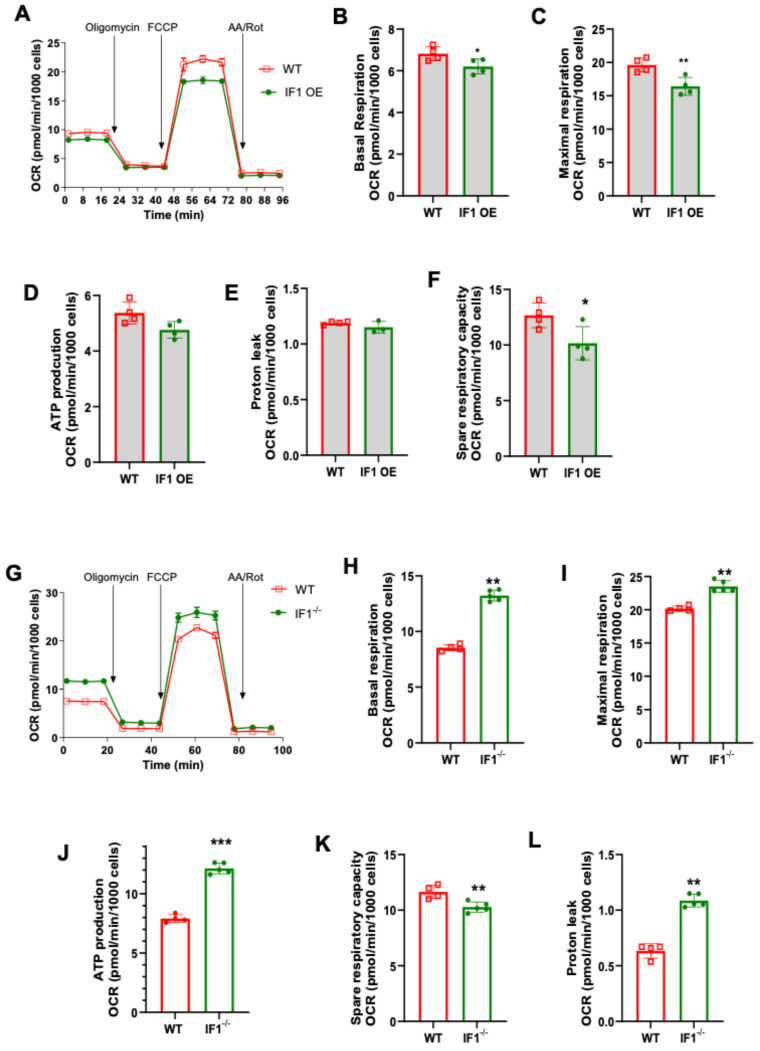
Effects of IF1 on cellular respiration in cultured IF1 OE, IF1^−/−^ and WT MEFs under normoxia. A mitochondrial stress test on cultured IF1^−/−^ and IF1 OE MEFs under normoxia. Mitochondrial oxygen consumption rate (OCR) representing cellular respiration in MEFs was determined using a Mitochondrial Stress Test and a Seahorse XFe24 Analyzer. (**A**) Overall trace of mitochondrial stress test on IF1 OE and WT MEFs. (**B**) Basal respiration. (**C**) Maximal respiration. (**D**) ATP production rate. (**E**) Spare respiratory capacity. (**F**) Proton leak. (**G**) Overall trace of mitochondrial stress test on IF1^−/−^ and WT MEFs. (**H**) Basal respiration. (**I**) Maximal respiration. (**J**) ATP production rate. (**K**) Spare respiratory capacity. (**L**) Proton leak. Data were obtained from *n* = 4 wells for each independent experimental group. Data are reported as mean ± SEM. Data were analyzed using Student’s *t*-test with Welch correction. * *p* < 0.05, ** *p* < 0.01, *** *p* < 0.001.

**Figure 5 cells-13-00551-f005:**
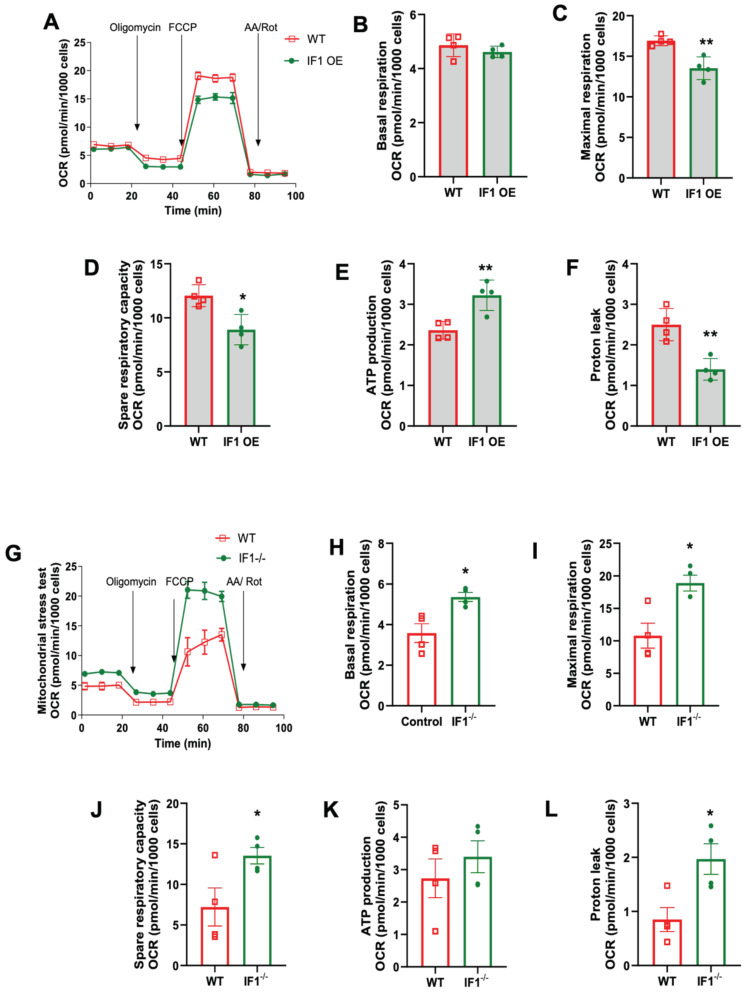
Effects of IF1 on cellular respiration in cultured MEFs subjected to hypoxia/reoxygenation. A mitochondrial stress test on cultured IF1^−/−^ and IF1 OE MEFs under the hypoxia. (**A**) Overall trace of mitochondrial stress test on IF1 OE and WT MEFs. (**B**) Basal respiration. (**C**) Maximal respiration. (**D**) ATP production rate. (**E**) Spare respiratory capacity. (**F**) Proton leak. (**G**) Overall trace of mitochondrial stress test on IF1^−/−^ and WT MEFs. (**H**) Basal respiration. (**I**) Maximal respiration. (**J**) ATP production rate. (**K**) Spare respiratory capacity. (**L**) Proton leak. Data were obtained from *n* = 4 wells for each independent experimental group. Data are reported as mean ± SEM. Data were analyzed using Student’s *t*-test with Welch correction. * *p* < 0.05, ** *p* < 0.01.

**Figure 6 cells-13-00551-f006:**
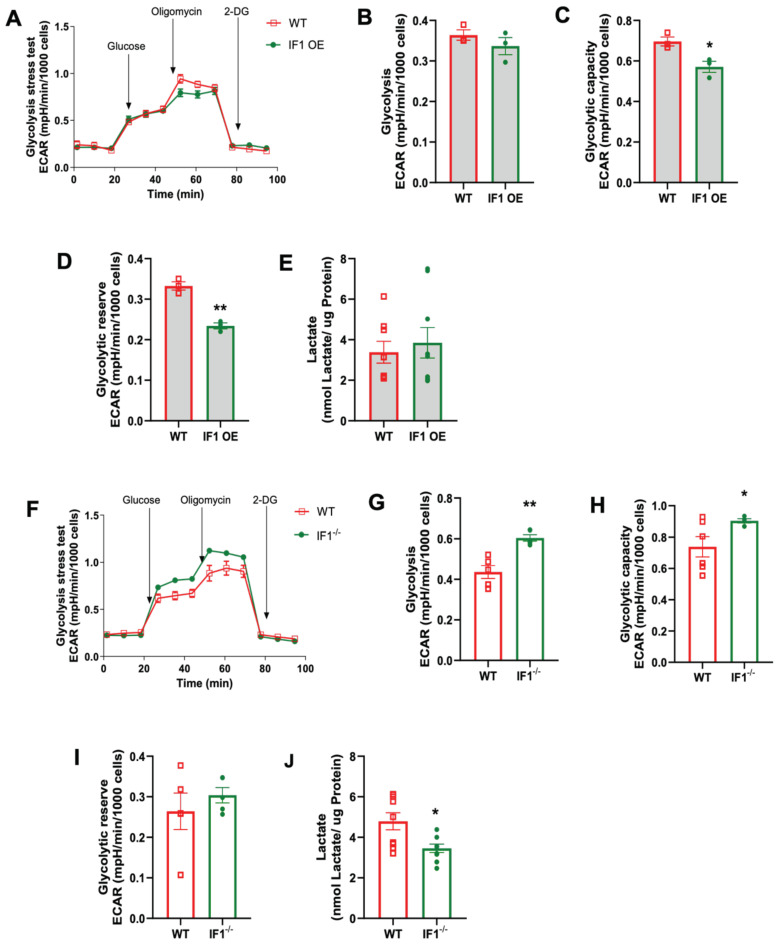
Effects of IF1 on cellular glycolysis in cultured IF1 OE, IF1^−/−^ and WT MEFs under normoxia. The extracellular acidification rate (ECAR) representing cellular glycolysis in MEFs was determined using a glycolysis stress test and a Seahorse XFe24 Analyzer. (**A**) Overall trace of ECAR of the glycolysis stress test on IF1 OE and WT MEFs under normoxia. (**B**–**D**) Glycolysis, glycolytic capacity, and glycolytic reserve in WT and IF1 OE MEFs under normoxia. (**E**) Lactate contents in WT and IF1 OE MEFs under normoxia. (**F**) Overall trace of ECAR of the glycolysis stress test on IF1^−/−^ and WT MEFs under normoxia. (**G**–**I**) Glycolysis, glycolytic capacity, and glycolytic reserve in WT and IF1^−/−^ MEFs under normoxia. (**J**) Lactate contents in WT and IF1^−/−^ MEFs under normoxia. Data are reported as mean ± SEM. Experiments were repeated 3 times with consistent results. Data were analyzed using Student’s *t*-test with Welch correction. * *p* < 0.05, ** *p* < 0.01.

**Figure 7 cells-13-00551-f007:**
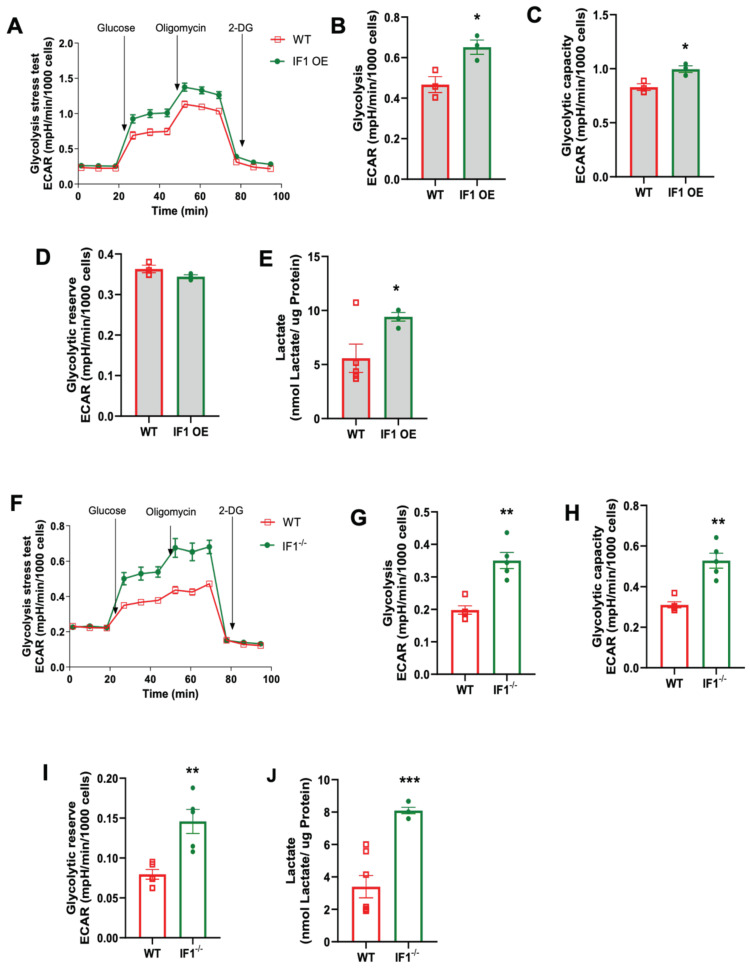
Effects of IF1 on cellular glycolysis in cultured IF1 OE, IF1^−/−^ and WT MEFs subjected to hypoxia/reoxygenation. The extracellular acidification rate (ECAR) representing cellular glycolysis in MEFs was determined using a glycolysis stress test and a Seahorse XFe24 Analyzer. (**A**) Overall trace of ECAR of the glycolysis stress test on IF1 OE and WT MEFs under hypoxia. (**B**–**D**) Glycolysis, glycolytic capacity, and glycolytic reserve in WT and IF1 OE MEFs under hypoxia. (**E**) Lactate contents in WT and IF1 OE MEFs under hypoxia. (**F**) Overall trace of ECAR of the glycolysis stress test on IF1^−/−^ and WT MEFs under hypoxia. (**G**–**I**) Glycolysis, glycolytic capacity, and glycolytic reserve in WT and IF1^−/−^ MEFs under hypoxia. (**J**) Lactate contents in WT and IF1^−/−^ MEFs under hypoxia. Data are reported as mean ± SEM. Experiments were repeated 3 times with consistent results. Data were analyzed using Student’s *t*-test with Welch correction. * *p* < 0.05, ** *p* < 0.01, *** *p* < 0.001.

## Data Availability

Data can be made available via contacting corresponding author.

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
