# Peer review of "IF1 Promotes Cellular Proliferation and Inhibits Oxidative Phosphorylation in Mouse Embryonic Fibroblasts under Normoxia and Hypoxia"

_cells, 2024, doi:10.3390/cells13060551_

Round 1
Reviewer 1 Report
Comments and Suggestions for Authors
In this manuscript, the authors claim that overexpression of IF1, an inhibitory subunit of mitochondrial ATP synthase, suppresses mitochondrial oxidative phosphorylation and preserves mitochondrial ATP. In contrast, cells lacking IF1 can significantly increase cellular respiration and glycolysis without causing mitochondrial ATP depletion. Although the results are potentially interesting, I have several points that should be addressed by the authors. The specific points are as follows.
Major points.
1. The amount of ATP synthesis and the proliferative capacity of the cells do not correspond at all and cannot be explained. First, overexpression of IF1 under normoxic conditions should theoretically decrease the amount of ATP, but in fact the amount of ATP is unchanged (Figure 3B) and the proliferative potential of the cells is still increased (Figure 1A). Also, in cells lacking IF1 under normoxic conditions, the amount of ATP should theoretically increase, but in fact the amount of ATP is unchanged (Figure 3B) and the proliferative capacity of the cells is still slightly increased (Figure 1B). Furthermore, the amount of ATP was dramatically increased only when IF1 was overexpressed under hypoxic conditions (Figure 3D), but under hypoxic conditions the proliferative capacity of the cells was markedly increased regardless of whether IF1 was present or not (Figure 2A and 2B). The authors need to rationally explain these phenomena; I am concerned that the experimental results for OCR and glycolytic capacity are only phenomenological and that the authors have not captured the essential cellular effects nor performed a detailed molecular biological analysis.
2. Figure 3: The authors claim "real-time" imaging, but they only provide an image at a single point in time. Changes over time should be shown. The microscopic image on the left (Figures 3A and 3C) does not match the values of the quantitative results on the right (Figures 3B and 3D). Quantification of microscopic images is not reliable, and analysis by flow cytometry or other methods is desirable.
Minor points.
1. English should be carefully revised throughout the manuscript.
2. The authors claim that IF1 expression is increased under hypoxic conditions, but how does this compare to IF1 overexpression under normoxic conditions? Also, how do expression levels change when hypoxia and IF1 overexpression are combined compared to each alone? It would be desirable to compare protein levels of all of these (normoxia, hypoxia, OE, and deficiency) simultaneously.
Author Response
Reviewer 1
We are grateful for your constructive critiques. Following are our responses to the concerns:
Major points.
- The amount of ATP synthesis and the proliferative capacity of the cells do not correspond at all and cannot be explained. First, overexpression of IF1 under normoxic conditions should theoretically decrease the amount of ATP, but in fact the amount of ATP is unchanged (Figure 3B) and the proliferative potential of the cells is still increased (Figure 1A). Also, in cells lacking IF1 under normoxic conditions, the amount of ATP should theoretically increase, but in fact the amount of ATP is unchanged (Figure 3B) and the proliferative capacity of the cells is still slightly increased (Figure 1B). Furthermore, the amount of ATP was dramatically increased only when IF1 was overexpressed under hypoxic conditions (Figure 3D), but under hypoxic conditions the proliferative capacity of the cells was markedly increased regardless of whether IF1 was present or not (Figure 2A and 2B). The authors need to rationally explain these phenomena; I am concerned that the experimental results for OCR and glycolytic capacity are only phenomenological and that the authors have not captured the essential cellular effects nor performed a detailed molecular biological analysis.
We are grateful for this reviewer to bring up the key arguments in the field regarding how IF1 may affect ATP synthase. It has been well documented that IF1 increases its affinity for ATP synthase under acidic conditions. In normoxia, IF1’s inhibitory effects on ATP synthase are therefore limited and exert minimal impacts on ATP synthesis (Figure 3B). However, the energetic levels of these cells seem sufficient to support cell proliferation (Figure A). For the same reason, MEFs lacking IF1 under normoxic conditions, ATP is unchanged (Figure 3B). However, the reason why the proliferation of IF1 KO cells is slightly increased is unclear. It is likely that IF1 changes in MEFs trigger other ATP synthase unrelated alterations that enhance cellular glycolysis and cellular proliferation.
The amount of mitochondrial ATP was dramatically increased only when IF1 was overexpressed under hypoxia due to increased IF1 interaction with and inhibition of ATP synthase activity in the acidic state and the increase in cellular glycolysis. Under hypoxic conditions the proliferative capacity of the cells was markedly elevated regardless of whether IF1 was present or not (Figure 2A and 2B). This phenomenon has been reported previously (Lee SH et al, Cell Prolif, 2008, 41(2): 230-247), thus it is likely an IF1-independent cell growth.
The experimental results of OCR and glycolytic capacity should not be substantially affected by cell proliferation and survival states. This is because we used the same number of cells for the experiment. It is reasonable to believe that the metabolic flux measurement would capture at least the major cellular effects of IF1.
- Figure 3: The authors claim "real-time" imaging, but they only provide an image at a single point in time. Changes over time should be shown. The microscopic image on the left (Figures 3A and 3C) does not match the values of the quantitative results on the right (Figures 3B and 3D). Quantification of microscopic images is not reliable, and analysis by flow cytometry or other methods is desirable.
Due to the subtle changes at earlier timepoints and the intense labor involved, we did not measure fluorescent intensity at over times via time lapse videos. The changes we quantified serve the purpose of detecting the main effects of IF1 within mitochondria in cultured MEFs under normoxic and hypoxic conditions. We thank the reviewer for pointing out the mismatched imaging. We have corrected this error.
Using a flow cytometry method is not feasible because the ATP indicator was not robustly and uniformly transduced in the MEFs. The visualization of ATP within mitochondria provides a more accurate measurement of mitochondrial ATP in live cells than most available methods that often have issues like low signal to noise ratio.
Minor points.
- English should be carefully revised throughout the manuscript.
Yes. English is carefully checked and revised by native speaking co-authors.
- The authors claim that IF1 expression is increased under hypoxic conditions, but how does this compare to IF1 overexpression under normoxic conditions? Also, how do expression levels change when hypoxia and IF1 overexpression are combined compared to each alone? It would be desirable to compare protein levels of all of these (normoxia, hypoxia, OE, and deficiency) simultaneously.
Yes. The IF1 transcript increased in MEFs subjected to short-term hypoxia. We do not have time to repeat whole experiment to gain cells and run WB due to the limited time for resubmission. However, we do not expect to see a major increase in IF1 protein in cells subjected to short-term (3 hours) hypoxia.

Reviewer 2 Report
Comments and Suggestions for Authors
Research paper “IF1 promotes cellular proliferation and suppresses oxidative phosphorylation in mouse embryonic fibroblasts under normoxia and hypoxia” by Lauterboeck et al. aims to explore the role of ATP synthase inhibitory factor 1 (ATP-IF1) in normoxia and hypoxia in primary mouse embryonic fibroblasts. This article lacks novelty other than the use of primary fibroblasts. The authors presented the observations but failed to present a new hypothesis and move the field forward. Below are some of the concerns.
- The authors did not mention whether the wild-type (WT) MEFs are littermate and sex-matched. It is essential to have the littermate control MEFs to normalize the effects of mitochondrial DNA variations. Similarly, the Y chromosome has genes associated with several biological processes, including metabolism. Therefore, it is essential to establish the littermate and sex-matched controls in the case of MEFs.
- The Author concludes that “IF1 exerts protective and proliferative actions, and IF1 knockout exerts a protective action on cultured MEFs when subjected to hypoxia.” this seems to be an over-interpretation of the presented data. Both knockout and overexpression of IF1 seem to increase proliferation regardless of oxygen levels with trypan blue exclusion assays. The rest of all the assays used to measure the proliferation show marginal changes. Also, the figure legend is unclear for BrdU incorporation and Ki67 staining assays. How many cell numbers per experiment were used to calculate the data plotted?
- The central premise of this study is to understand the role of IF1 in normoxia and hypoxia in primary cells. However, the authors failed to highlight the effect of hypoxia on the normal cells while presenting the data. For example, in Figure 3, the authors conclude that IF1 preserves mitochondria ATP in hypoxia without showing the effect of hypoxia on mitochondrial ATP in WT MEFs. This is important to establish whether IF1 has any protective role of mitochondrial ATP.
- Like Figure 1, authors have overinterpreted the results of Figures 4 and 5. When there is no data in the manuscript about reoxygenation, why do the authors conclude that “IF1 impacts cellular respiration in normoxia and hypoxia/reoxygenation under cultured conditions.”
- The authors mentioned observations for the IF1 knockout and overexpression mouse in the discussion section without presenting the relevant data. At this time, such practices should be discouraged. Also, the discussion reads more of reciting the data instead of discussing and presenting a new hypothesis.
Author Response
Reviewer 2
We are grateful for your constructive critiques. Following are our responses to the concerns:
- The authors did not mention whether the wild-type (WT) MEFs are littermate and sex-matched. It is essential to have the littermate control MEFs to normalize the effects of mitochondrial DNA variations. Similarly, the Y chromosome has genes associated with several biological processes, including metabolism. Therefore, it is essential to establish the littermate and sex-matched controls in the case of MEFs.
We agree. We could have used littermate MEFs as control. We compared IF1 OE MEFs with littermate MEFs in our early experiments. We did not detect any major differences. As a result of a relocation incident, we lost those MEFs. All MEFs were prepared with identical preparation with the same passages between IF1 ablations and WT. The concern related to sex differences is interesting, but we found it challenging to separate male and female among pups at embryonic day 13.5 when generating MEFs.
- The Author concludes that “IF1 exerts protective and proliferative actions, and IF1 knockout exerts a protective action on cultured MEFs when subjected to hypoxia.” this seems to be an over-interpretation of the presented data. Both knockout and overexpression of IF1 seem to increase proliferation regardless of oxygen levels with trypan blue exclusion assays. The rest of all the assays used to measure the proliferation show marginal changes. Also, the figure legend is unclear for BrdU incorporation and Ki67 staining assays. How many cell numbers per experiment were used to calculate the data plotted?
To address the over-interpretation concern of this reviewer, we rewrote the sentence to avoid that.
We also clarified the experimental details by adding the following in the Figure legends: “For image data analysis, four fields per well were taken, and at least 20 cells per field were counted for colocalization of either BrdU or KI-67 with Hoechst” to clarify how many cells per experiment were used to calculate the data plotted.
- The central premise of this study is to understand the role of IF1 in normoxia and hypoxia in primary cells. However, the authors failed to highlight the effect of hypoxia on the normal cells while presenting the data. For example, in Figure 3, the authors conclude that IF1 preserves mitochondria ATP in hypoxia without showing the effect of hypoxia on mitochondrial ATP in WT MEFs. This is important to establish whether IF1 has any protective role of mitochondrial ATP.
Thank you for pointing out the error. We added WT image in hypoxia to indicate that IF1 preserved mitochondrial ATP.
- Like Figure 1, authors have overinterpreted the results of Figures 4 and 5. When there is no data in the manuscript about reoxygenation, why do the authors conclude that “IF1 impacts cellular respiration in normoxia and hypoxia/reoxygenation under cultured conditions.”
Sorry for the confusion. We corrected the title of Figure 5 and Figure 7 to clarify that those experiments were from MEFs subjected to hypoxia/reoxygenation.
- The authors mentioned observations for the IF1 knockout and overexpression mouse in the discussion section without presenting the relevant data. At this time, such practices should be discouraged. Also, the discussion reads more of reciting the data instead of discussing and presenting a new hypothesis.
To address this concern, we revised the discussion section to minimize irrelevant statements and reciting data.

Reviewer 3 Report
Comments and Suggestions for Authors
This paper by Lauterboeck et al Yang, makes an extensive SeaHorse study on the effects of IF1 deletion and/or overexpression on the cell physiology, mitochondrial bioenergetics parameters and cell glucolytic metabolism. Althought most of the results are properly presented, some the conclusions are not sustained by the data as presented, particularly regarding the conclusion that IF1 inhibits the ATP synthase, since in none of their dfferent analyses is clearly shown that IF1 overexpression inhibits the ATP synthase turnover. Rather, their results colectively show that IF1 inhibits the F-ATPase activity and thus preserves mitochodrial ATP, thus their data support the recent proposals suggesting that IF1 inhibits exclusively the "reverse" F-ATPase activity but not the "forward" ATP synthase rotation. Some WB analyss are nor porperly presented and the full discussion must be improved in a major revision that could be further considered for publication in this journal.
Detalied comments:
1. In the abstract remove in line 33 the text concluding that IF1 inhibits OXPHOS and the ATP synthase, and replace it by saying: "Therefore, we conclude that IF1 mainly inhibits the mitochondrial F1FO-ATPase activity, but not the ATP synthase turnover".
2. In line 44 replace "catalyze" with "convert"
3. In line 73 inside the parenthesis of citation (11-13), and before citation 11, insert the mising citation that demonstrated firstly the higher [IF1] contents in cancer cells: Bravo C etal García JJ J Bioenerg. Biomembr 36(3):257-267 (2004).
4. Regarding the supplemenatary Figure 1. The MW of the mature Mus musculus IF1 is 9.5 kDa (this can be quickly checked in Expasy Protparam), but not 12kDa as indicated in Suppl Fig 1A, mature mouse IF1 should co-migrate with the standard of 10 kDa, or slightly below 10 kDa, but in their WBs the apparent IF1 band runs in ≈12 kDa, that is too high for Mus musculus IF1. Coincidentally, the MW of the mitochondrial precursor mouse IF1 is 12.1 kDa, so it is possible that the protein that was detected in their WBs is the mitochondrial precursor of Mus musculus IF1, but not the mature protein. However, in such case, the WB should show two IF1 bands on each lane, one for the mitochondrial IF1 precursor of ≈ 12kDa, and another of he mature IF1 of ≈ 10 kDa.Additionally, the source of the Ab anti-IF1 is not detailed in methods or in Supp Material, so which is the source of the anti-IF1 Ab? is it monoclonal or polyclonal? if this is polyclonal this could be detecting something else of 12 kDa that is not the real IF1 band (probably the IF1 precursor as indicaed or something else). This a major drawback, the authors must show better anti-IF1 WBs in a revised version showing that the IF1 band run closer to 10 kDa, otherwise the full paper falls into disbelief. The authors should also explain the source (commercial company and name) of the anti-IF1 antibody. The dilution used of this anti-IF1 antibody in the WB must be also explained. This is because even some monoclonals anti-IF1 sometimes give an additional non-specific band above IF1, which is not IF1, so it is important to describe de Ab source in detail. In addition, a load control (actin or whatever constant protein is also missing in Suppl Fig 1A, but for some reason these loading controls are actually shown in Supp Fig2, but why not in Supp Fig1A?). In addition, the MW of the markers in WB of figure 1A must be indicated as well as in the future requested improved and revised anti-IF1 WB analyses. One can guess that the lower marker is 10 kDa, and the upper marker is 15 kDa, so the apparent size of the apparent IF1 band gives 12 kDa, but the actual sizes of the markers are not porperly indicated, these should be clearly shown in the revised version of this figure and manuscript. Furhermore, the SDS-PAGE technique is nor properly described in materials amd methods, in the WB methods, the SDS-PAGE is described as pre-casted BioRad gels, but it is not clarified whether the SDS-PAGE gels were ran according to the Tris/Glycine Laemmli protocol, or following the Tris-Tricine procedure of Schägger and Von-Jagow , this should also be described in detail in Methods adding the respective citation in the text and its reference in the reference list, either that of Laemmli or the one of Schägger and Von-Jagow. Other important details are that the WBs should indicate how many ug of total protein were loaded on each lane, and the commercial source and name of the MW standards. The revised version of this paper should indicate all of this important information in order to decide if the quality and feasibility of this important WB anti IF1 is trustable. Currently, as presented this WB anti-F1 cannot provide trustable information on the presence or absence, or on relative amounts of IF1 in the different samples loaded in this important WB. It is also higly recommended to load one lane of this WB anti-IF1 with the recombinant purified mouse IF1, in order to confirm that the apparent IF1 band developed is actually the right mouse IF1 protein. It is also suggested that the full WB image should be presented since it is in Supplementary material, and not only the partial image segment of the apparent IF1 band and standards. This is because a full cell extract should give some non-specific protein bands above and below IF1, so it is better in this case to show the full WB image and not only the apparent IF1 band segment, the inclusion of a lane containing the purified recombinant mouse IF1 should give the specific band without further non specific bands. FInally, the method to obtain the cell extract to load the SDS-PAGE gel and WB is also missing in Materials and Methods. It is also important that the authors must explain in detail how they obtained the cell extract to load the lanes. How many cells where used to obtain the extract? was it sonicated or not? was it centrifuged before loading he gels or not? Did they measured, as they had should, the total extract protein concentration before loading the gel? which method was used to measure the final protein concentration of the cell extracts? how may micrograms of protein were loaded on each lane? All this important missing information must be detailed in materials and methods or Supplementary material in the revised version of this paper, the authors should recall the the methods must be fully detailed in order to allow any other laboratory in the world to reproduce their experiments in full detail, so it is impossible for any reader to repeat these experiments without knowing all these important details.
5. In line 372 insert references: 1) Jackson and Harris, FEBS Lett 229: 224 (1988); Mimura et al Tagawa J biochem (Tokyo) 113:350 (1993); Minauro-Sanmiguel et al García JJ J Bioenerg. Biomembr. 34(6):433-443 (2002). Ref 3 (Cabezón etal 2003) instead of ref. 25-27. These are the original papers demonstrating the isnertion of IF1 into the alfaDP,BetaDP, gamma interface of IF1. The indicated references 25-27 are only reviews that have little to do with the demonstration of the inhibitory binding site of IF1. Insert the missing citatons in the reference list.
6. In line 392 Insert: "How removal and/or overexpression of IF1 lead to the same pro-survival effects is an intriguing question. The same cellular effects of removal or OE of IF1 might be associated to compensatory effects after IF1 genetic removal. For instance, it has been discussed that some IF1 homologues, isoforms, or gene copies may exist in eukaryotes from yest to animals and humans (Mendoza-Hoffmann et al J Bioenerg Biomembr 50(5):403-424 (2018)). Some IF1 sioforms or gene copies in mouse can be found in NCBI (https://www.ncbi.nlm.nih.gov/ipg/NP_001407692.1). Therefore, removal of a single IF1 gene as made in this study, may lead to compensatory expression of other IF1 copies or isoform genes that may be present in the eukaryotic chromosomes, this could explain why removal or overexpression of IF1 may lead to a similar phenotype in whole cells".
7. In line 393. The discussion lacks the well known effects of IF1 as dimerizing and oligomerizing factor of the mitochondrial ATP synthase, inlcuding the main citations and references in this regard. This discussion is suggested to be inserted or modified as follows after the word "understood" in line 393: It has been shown that IF1 is involved in regulating mitochondrial morphology through cristae
formation by IF1 stabilization of the dimeric and oligomeric ATP synthases (García JJ et al, Biochemistry, 45:12695-12703 (2006); MInauro Sanmiguel et al, PNAS 102(35):12356-12358 (2005); Gu J et al... Yang M Science 364: 1068-1075 (2019) so these pro-cristae formation effects of IF1 may contribute to its pro-survival effects . On the other hand, it has been suggested that IF1 may participate in mitochondrial cristae density and mitophagy via ....etc.
8. In line 405 remove "y" and replace with "i", it should read "Oligomycin"
9. In line 414 remove text repetititive text indicated ("whereas the IF1 rposurvival effects are likely via"), it should read "...IF1 OE are via ATP preservation and improved cellular respiration."
10. In line 419, remove "and ATP production-linked OCR".
In hypoxia, IF1 OE decreased mitochondrial respiration, but ATP production was increased instead of decreased (Fig. 5K), so there is no decrease in ATP production linked to IF1 OE in hypoxia.
11. In line 440, Replace "ATP synthase" with "F1FO-ATPase" Once more because there is no evidence showing that IF1 inhibits the ATP synthase activity, but only the F-ATPase is inhibited.
12. Replace text in line 445 and emove text in lines 446-448. Remove "hyper" and insert "de" it should read "depolarization". This is according to the red bars in Supplementary figure 3B, the IF1 OE DECREASES the red fluorescence of TMRM, this means that excess IF1 DEPOLARIZES the mitochondrial membrane, but not hyperpolarizes it. Thus the discussion of lines 446-448 are non sense, once more, the actual effect of IF1 on membrane potential was the opposite (depolarization) rather than hyperpolarization, so these results do not support the concluding inference that IF1 inhibits the ATP synthase activity, so remove these text lines.
13. In line 454 Replace "OXPHOS" with "respiration". It should read: "Meanwhile the reduced respiration rate in MEFs..."
This is because in Fig 4A and 4D, respiration is clearly decreased but ATP production is not signiflcantly affected, thus the overall OXPHOS is not decreased, but only respiration in normoxia.
14. In lines 459-463, Remove all the text in the the lines 459-463 starting with removing the phrase "and synthesis", and replace it with "but there is no evidence showing that IF1 actually inhibits the forward ATP synthesis turnover. This is in concordance with recent proposals suggesting that mitochondrial IF1, bacterial ε and the α-proteobacterial ζ subnit, all of these ATP synthase inhibitors bind into the same αDP/βDP/γ interface and work as pawl-ratchets or unidirectional inhibitors by blocking exclusively the "backward" F1FO-ATPase rotation, but not the "forward" F1FO-ATP synthase turnover (Tsunoda et al, Proc. Natl. Acad. Sci. USA 98, 6560–6564(2001); Garcia-Trejo etal, J Biol Chem 291(2):538-546 (2016); Mendoza-Hofmann et al Cell Rep 22(4):1067-1078 (2018); Mendoza-Hoffmann et al J Bioenerg. Biomembr 50(5):403-424 (2018)"
This insertion is important because in none of these experiments it was shown that ATP production was inhibited significantly by IF1 overexpression (see for instance Figure 4D where the apparent decrease in ATP production associated with IF1 OE in normoxia is not statistically significant). In contrast, it was found that IF1 OE increased ATP production under hypoxia (Fig 5K). Besides the increase in ATP production after removal of IF1 (IF1-/-) parallels the incease in respiration, so the increase in ATP after IF1 removal in hypoxia (Fig 5J) can be explained by a higher respiration (Fig 5A) and membrane potential fueling the ATP synthase, rather than by "unlocking" the ATP synthase turnover by IF1 removal. Furthermore, It is clear that IF1 removal decreases the full mitochondrial ATP content and IF1 OE increases mitochondrial ATP content (Figure 3), so the combined data show that IF1 preserves mitochondrial ATP by inhibiting its hydrolsis, but in none of the experiments is shown that IF1 inhibits ATP synthesis carried out by the mitohonrial ATP synthase.
15. In line 465 Insert missing citation: Cortés-Hernández et al Biochem. Biophys. Res. Commun. (BBRC) 330: 844-849 (2005). This paper showed for the first time the association of IF1 with the plasma membrane ATP synthase in the endothelial cells surface. This citation should be inserted before ceitation of ref 37 inside the parenthesis (37-40), and the reference must be cited prooerly in the reference list.
16. The conclusions are fine since these do no state that IF1 inhibits the ATP synthase, but that influcences cell viability, and oxidative state and glycolysis, only the word "state" is missing in line 472. So in this line insert: "state", it should read "...oxidative state and glycolysis"
All these points must be properly addressed in order to consider a revised version for publication.
Comments on the Quality of English Language
The quality of english can be improved, some suggestions are indicated in the suggested corrections in the text.
Author Response
Reviewer 3
We are grateful for your constructive critiques. Following are our responses to the concerns.
This paper by Lauterboeck et al Yang, makes an extensive SeaHorse study on the effects of IF1 deletion and/or overexpression on the cell physiology, mitochondrial bioenergetics parameters and cell glucolytic metabolism. Althought most of the results are properly presented, some the conclusions are not sustained by the data as presented, particularly regarding the conclusion that IF1 inhibits the ATP synthase, since in none of their dfferent analyses is clearly shown that IF1 overexpression inhibits the ATP synthase turnover. Rather, their results colectively show that IF1 inhibits the F-ATPase activity and thus preserves mitochodrial ATP, thus their data support the recent proposals suggesting that IF1 inhibits exclusively the "reverse" F-ATPase activity but not the "forward" ATP synthase rotation. Some WB analyss are nor porperly presented and the full discussion must be improved in a major revision that could be further considered for publication in this journal.
We may incompletely understand this concern. However, we would like to point out that the investigation is not attempt to assess ATP synthase turnover. The main goal of the investigation is to determine how IF1 impacts cellular respiration and glycolysis, cellular proliferation and survival and the mitochondrial ATP content as the end results of the above changes. Based on the above outcomes, we could only indirectly propose if IF1 inhibits unidirectional or bidirectional activities of ATP synthase. We revised the discussion to acknowledge this point.
Some WB analyss are nor porperly presented and the full discussion must be improved in a major revision that could be further considered for publication in this journal.
To address the concern regarding the WB analysis, we now provided details in the method section.
Detalied comments:
- In the abstract remove in line 33 the text concluding that IF1 inhibits OXPHOS and the ATP synthase, and replace it by saying: "Therefore, we conclude that IF1 mainly inhibits the mitochondrial F1FO-ATPase activity, but not the ATP synthase turnover".
Because our investigation did not assess ATP synthase turnover, we cannot draw any conclusion related to ATP synthase turnover.
- In line 44 replace "catalyze" with "convert"
As suggested, we made the change.
- In line 73 inside the parenthesis of citation (11-13), and before citation 11, insert the mising citation that demonstrated firstly the higher [IF1] contents in cancer cells: Bravo C etal García JJ J Bioenerg. Biomembr 36(3):257-267 (2004).
As suggested, we made the change.
- Regarding the supplemenatary Figure 1. The MW of the mature Mus musculus IF1 is 9.5 kDa (this can be quickly checked in Expasy Protparam), but not 12kDa as indicated in Suppl Fig 1A, mature mouse IF1 should co-migrate with the standard of 10 kDa, or slightly below 10 kDa, but in their WBs the apparent IF1 band runs in ≈12 kDa, that is too high for Mus musculus IF1. Coincidentally, the MW of the mitochondrial precursor mouse IF1 is 12.1 kDa, so it is possible that the protein that was detected in their WBs is the mitochondrial precursor of Mus musculus IF1, but not the mature protein. However, in such case, the WB should show two IF1 bands on each lane, one for the mitochondrial IF1 precursor of ≈ 12kDa, and another of he mature IF1 of ≈ 10 kDa.Additionally, the source of the Ab anti-IF1 is not detailed in methods or in Supp Material, so which is the source of the anti-IF1 Ab? is it monoclonal or polyclonal? if this is polyclonal this could be detecting something else of 12 kDa that is not the real IF1 band (probably the IF1 precursor as indicaed or something else). This a major drawback, the authors must show better anti-IF1 WBs in a revised version showing that the IF1 band run closer to 10 kDa, otherwise the full paper falls into disbelief. The authors should also explain the source (commercial company and name) of the anti-IF1 antibody. The dilution used of this anti-IF1 antibody in the WB must be also explained. This is because even some monoclonals anti-IF1 sometimes give an additional non-specific band above IF1, which is not IF1, so it is important to describe de Ab source in detail. In addition, a load control (actin or whatever constant protein is also missing in Suppl Fig 1A, but for some reason these loading controls are actually shown in Supp Fig2, but why not in Supp Fig1A?). In addition, the MW of the markers in WB of figure 1A must be indicated as well as in the future requested improved and revised anti-IF1 WB analyses. One can guess that the lower marker is 10 kDa, and the upper marker is 15 kDa, so the apparent size of the apparent IF1 band gives 12 kDa, but the actual sizes of the markers are not porperly indicated, these should be clearly shown in the revised version of this figure and manuscript. Furhermore, the SDS-PAGE technique is nor properly described in materials amd methods, in the WB methods, the SDS-PAGE is described as pre-casted BioRad gels, but it is not clarified whether the SDS-PAGE gels were ran according to the Tris/Glycine Laemmli protocol, or following the Tris-Tricine procedure of Schägger and Von-Jagow , this should also be described in detail in Methods adding the respective citation in the text and its reference in the reference list, either that of Laemmli or the one of Schägger and Von-Jagow. Other important details are that the WBs should indicate how many ug of total protein were loaded on each lane, and the commercial source and name of the MW standards. The revised version of this paper should indicate all of this important information in order to decide if the quality and feasibility of this important WB anti IF1 is trustable. Currently, as presented this WB anti-F1 cannot provide trustable information on the presence or absence, or on relative amounts of IF1 in the different samples loaded in this important WB. It is also higly recommended to load one lane of this WB anti-IF1 with the recombinant purified mouse IF1, in order to confirm that the apparent IF1 band developed is actually the right mouse IF1 protein. It is also suggested that the full WB image should be presented since it is in Supplementary material, and not only the partial image segment of the apparent IF1 band and standards. This is because a full cell extract should give some non-specific protein bands above and below IF1, so it is better in this case to show the full WB image and not only the apparent IF1 band segment, the inclusion of a lane containing the purified recombinant mouse IF1 should give the specific band without further non specific bands.
Thank you for the suggestion. We added the source, name, clone and used concentration of the IF1 antibody to the supplementary section of methods.
The mobility of the protein in the PAGE gel may not reflect the theoretical molecular weight. We have tested at least 5 different anti-IF1 antibodies and many of them detected a 14 KD band of an unknown protein. This band must not be IF1 because this band shown in WB with samples extracted from our IF1 KO and not overlapped with the overexpressed IF1 band in overexpression mice. Many previous publications also validated the 12 KD IF1 band. We have previously described IF1 from tissues of these mouse lines in peer-reviewed publications (Yang, K et al and Zhang K et al).
FInally, the method to obtain the cell extract to load the SDS-PAGE gel and WB is also missing in Materials and Methods. It is also important that the authors must explain in detail how they obtained the cell extract to load the lanes. How many cells where used to obtain the extract? was it sonicated or not? was it centrifuged before loading he gels or not? Did they measured, as they had should, the total extract protein concentration before loading the gel? which method was used to measure the final protein concentration of the cell extracts? how may micrograms of protein were loaded on each lane? All this important missing information must be detailed in materials and methods or Supplementary material in the revised version of this paper, the authors should recall the the methods must be fully detailed in order to allow any other laboratory in the world to reproduce their experiments in full detail, so it is impossible for any reader to repeat these experiments without knowing all these important details.
To address the above concerns, we did our best to provide all the related experimental details in the revised manuscript.
- In line 372 insert references: 1) Jackson and Harris, FEBS Lett 229: 224 (1988); Mimura et al Tagawa J biochem (Tokyo) 113:350 (1993); Minauro-Sanmiguel et al García JJ J Bioenerg. Biomembr. 34(6):433-443 (2002). Ref 3 (Cabezón etal 2003) instead of ref. 25-27. These are the original papers demonstrating the isnertion of IF1 into the alfaDP,BetaDP, gamma interface of IF1. The indicated references 25-27 are only reviews that have little to do with the demonstration of the inhibitory binding site of IF1. Insert the missing citatons in the reference list.
Thank you for this valuable information and it has been incorporated into the discussion as suggested.
- In line 392 Insert: "How removal and/or overexpression of IF1 lead to the same pro-survival effects is an intriguing question. The same cellular effects of removal or OE of IF1 might be associated to compensatory effects after IF1 genetic removal. For instance, it has been discussed that some IF1 homologues, isoforms, or gene copies may exist in eukaryotes from yest to animals and humans (Mendoza-Hoffmann et al J Bioenerg Biomembr 50(5):403-424 (2018)). Some IF1 sioforms or gene copies in mouse can be found in NCBI (https://www.ncbi.nlm.nih.gov/ipg/NP_001407692.1). Therefore, removal of a single IF1 gene as made in this study, may lead to compensatory expression of other IF1 copies or isoform genes that may be present in the eukaryotic chromosomes, this could explain why removal or overexpression of IF1 may lead to a similar phenotype in whole cells".
We thank the reviewer for this suggestion and added the relevant discussion and references in the revised manuscript.
- In line 393. The discussion lacks the well known effects of IF1 as dimerizing and oligomerizing factor of the mitochondrial ATP synthase, inlcuding the main citations and references in this regard. This discussion is suggested to be inserted or modified as follows after the word "understood" in line 393: It has been shown that IF1 is involved in regulating mitochondrial morphology through cristae
formation by IF1 stabilization of the dimeric and oligomeric ATP synthases (García JJ et al, Biochemistry, 45:12695-12703 (2006); MInauro Sanmiguel et al, PNAS 102(35):12356-12358 (2005); Gu J et al... Yang M Science 364: 1068-1075 (2019) so these pro-cristae formation effects of IF1 may contribute to its pro-survival effects . On the other hand, it has been suggested that IF1 may participate in mitochondrial cristae density and mitophagy via ....etc.
We thank the reviewer for this suggestion and added the relevant discussion and references in the revised manuscript.
In line 405 remove "y" and replace with "i", it should read "Oligomycin"
Corrected. Thanks!
- In line 414 remove text repetititive text indicated ("whereas the IF1 rposurvival effects are likely via"), it should read "...IF1 OE are via ATP preservation and improved cellular respiration."
Changed as suggested.
- In line 419, remove "and ATP production-linked OCR".
In hypoxia, IF1 OE decreased mitochondrial respiration, but ATP production was increased instead of decreased (Fig. 5K), so there is no decrease in ATP production linked to IF1 OE in hypoxia.
These texts were removed.
- In line 440, Replace "ATP synthase" with "F1FO-ATPase" Once more because there is no evidence showing that IF1 inhibits the ATP synthase activity, but only the F-ATPase is inhibited.
- Replace text in line 445 and emove text in lines 446-448. Remove "hyper" and insert "de" it should read "depolarization". This is according to the red bars in Supplementary figure 3B, the IF1 OE DECREASES the red fluorescence of TMRM, this means that excess IF1 DEPOLARIZES the mitochondrial membrane, but not hyperpolarizes it. Thus the discussion of lines 446-448 are non sense, once more, the actual effect of IF1 on membrane potential was the opposite (depolarization) rather than hyperpolarization, so these results do not support the concluding inference that IF1 inhibits the ATP synthase activity, so remove these text lines.
We disagree with the above comment in 11 and 12. Our results based on the quantitative outcome of TMRM staining support that the mitochondrial membrane potential in IF1 OE MEFs was modestly increased. This result implicates that IF1 inhibits ATP synthase and cellular respiration unless future evidence supports a role of IF1 on upstream OXPHOS.
- In line 454 Replace "OXPHOS" with "respiration". It should read: "Meanwhile the reduced respiration rate in MEFs..."
This is because in Fig 4A and 4D, respiration is clearly decreased but ATP production is not signiflcantly affected, thus the overall OXPHOS is not decreased, but only respiration in normoxia.
We changed the texts as suggested.
- In lines 459-463, Remove all the text in the the lines 459-463 starting with removing the phrase "and synthesis", and replace it with "but there is no evidence showing that IF1 actually inhibits the forward ATP synthesis turnover. This is in concordance with recent proposals suggesting that mitochondrial IF1, bacterial ε and the α-proteobacterial ζ subnit, all of these ATP synthase inhibitors bind into the same αDP/βDP/γ interface and work as pawl-ratchets or unidirectional inhibitors by blocking exclusively the "backward" F1FO-ATPase rotation, but not the "forward" F1FO-ATP synthase turnover (Tsunoda et al, Proc. Natl. Acad. Sci. USA 98, 6560–6564(2001); Garcia-Trejo etal, J Biol Chem 291(2):538-546 (2016); Mendoza-Hofmann et al Cell Rep 22(4):1067-1078 (2018); Mendoza-Hoffmann et al J Bioenerg. Biomembr 50(5):403-424 (2018)" This insertion is important because in none of these experiments it was shown that ATP production was inhibited significantly by IF1 overexpression (see for instance Figure 4D where the apparent decrease in ATP production associated with IF1 OE in normoxia is not statistically significant). In contrast, it was found that IF1 OE increased ATP production under hypoxia (Fig 5K). Besides the increase in ATP production after removal of IF1 (IF1-/-) parallels the incease in respiration, so the increase in ATP after IF1 removal in hypoxia (Fig 5J) can be explained by a higher respiration (Fig 5A) and membrane potential fueling the ATP synthase, rather than by "unlocking" the ATP synthase turnover by IF1 removal. Furthermore, It is clear that IF1 removal decreases the full mitochondrial ATP content and IF1 OE increases mitochondrial ATP content (Figure 3), so the combined data show that IF1 preserves mitochondrial ATP by inhibiting its hydrolsis, but in none of the experiments is shown that IF1 inhibits ATP synthesis carried out by the mitohonrial ATP synthase.
We respectfully disagree with the above statement regarding the interpretation of our experiments. We had no evidence to support that IF1 actually inhibits the forward ATP synthesis turnover. Our results provide strong evidence regarding how cellular respiration is response to IF1 KO and IF1 overexpression. In fact, the overall changes point to a conclusion, although indirectly, that IF1 may inhibit both ATP hydrolysis and synthesis. Because our study did not assess other ATP synthase component proteins, we did not include the above references to overinterpret our experimental results.
In line 465 Insert missing citation: Cortés-Hernández et al Biochem. Biophys. Res. Commun. (BBRC) 330: 844-849 (2005). This paper showed for the first time the association of IF1 with the plasma membrane ATP synthase in the endothelial cells surface. This citation should be inserted before ceitation of ref 37 inside the parenthesis (37-40), and the reference must be cited prooerly in the reference list.
Added as suggested.
- The conclusions are fine since these do no state that IF1 inhibits the ATP synthase, but that influcences cell viability, and oxidative state and glycolysis, only the word "state" is missing in line 472. So in this line insert: "state", it should read "...oxidative state and glycolysis"
Revised as suggested.

Round 2
Reviewer 1 Report
Comments and Suggestions for Authors
The authors have addressed almost all of my concerns, and I have no further points.
Author Response
Thanks again for the constructive review and the suggests for improving the manuscript.
Reviewer 3 Report
Comments and Suggestions for Authors
There is a wrongly written couple of bar labels in Supplementary figure S3B, wrongly saying that Hypoxia and Normoxia conditions are applied to both couple of bars in figure S3B. I marked the labeling error in a red rectangle in both the original and new figures S3B (SEE ATTACHED IMAGES). It should read as in figure S3C where only one label either normoxyc or hypoxic conditions are indicated. In my original review, this labeling error lead me to wrongly see that the IF1 overexpression showed a decrease in membrane potential (red bars), because of the wrongly labeled bars names in figure S3C. On the other hand, there is indeed a drastic change in the TMRM fluorescence panels, the original figures had less contrast and showed little changes in the three panels (WT, IF1 OE, and IF1-/-). (SEE ATTACHED IMAGES) In high contrast, the new figure S3 shows a higher contrast and there are more clear changes in the same panels, which are more consistent with the statistical analyses of figures S3B and S3C. However, the drastic change between the original and new fluorescence images in figure S3 was not declared properly by the authors, the authors should have indicated that they changed the fluorescence images of these fig S3 panels, but for some reason they did not declared it. One had to look much deeper in these non-declared differences in Suppl data presentation. More importantly, the suppossed differences in membrane potential (TMRM fluorescence of figure S3B), which is concluded as indicating a modest increase in membrane potential, is not really clear given the data dispersion specially in IF1 Overexpressed samples (IF1OE green bars). And this is the only single data that the authors use to conclude in the paper´s Title, Abstact, and Discussion that IF1 inhibits the ATP synthase, but that is totally indirect, poor, and non-concluding evidence that IF1 inhibits the ATP synthase.
Therefore, I would strongly suggest that the sentences where it is indicated that IF1 inhibits OxPhos and the ATP synthase should be removed from the paper.
Some sentences should be removed. These are in lines 2 and 3 of the Title; lines 32, 33, and 34 of the Abstract; lines 456, 457, 458, 464, 475, 476, and 477 of the Disscussion.
Finally the wrong labelling of Suppl Figure S3B should be corrected indicating correctly the Hypoxia and Normoxia conditions as in figure S3C, otherwise the readers might be confused as I was during my deep review.

Author Response
Thanks again for this reviewer's careful review.
1) Reviewer comment: The changes in S3B was not declared properly by the authors.
Response: Sorry for this mistake. Yes. We did find we mixed up the images of S3B in our original submission. We corrected that in the last revision. We should have specified that in our revised response.
2) Reviewer comment: this is the only single data that the authors use to conclude in the paper´s Title, Abstact, and Discussion that IF1 inhibits the ATP synthase, but that is totally indirect, poor, and non-concluding evidence that IF1 inhibits the ATP synthase.
Response: As we stated in the manuscript, IF1’s role in inhibiting ATP synthase has been well-documented since its discovery 60 years ago. The remaining argument is whether it unilaterally inhibits hydrolytic or both bidirectionally synthetic and hydrolytic activities. Our in vivo cellular studies may indirectly support the latter, but as this reviewer pointed out, without direct evidence. The changes of cellular respiration in the IF1 ablated cells could be affected by other factors. Therefore, we revised the manuscript to avoid giving readers the impression that our present study is about to prove that IF1 directly inhibits ATP synthase and oxidative phosphorylation. We changes revised the manuscript accordingly with highlight.